# The Potential Function of *SiLOX4* on Millet Discoloration during Storage in Foxtail Millet

Qi Ma [1], Junjie Wang [1], Lu Cheng [1], Yaqiong Li [1,2], Qianxiang Zhang [1], Hongying Li [1,2,3], Yuanhuai Han [1,2,3], Xiaoxi Zhen [3,*] and Bin Zhang [1,2,*]

1   College of Agriculture, Shanxi Agricultural University, Jinzhong 030801, China
2   Institute of Agricultural Bioengineering, Shanxi Agricultural University, Jinzhong 030801, China
3   Shanxi Key Laboratory of Minor Crop Germplasm Innovation and Molecular Breeding,
    Shanxi Agricultural University, Taiyuan 030031, China
*   Correspondence: xiaoxizhen@sxau.edu.cn (X.Z.); binzhang@sxau.edu.cn (B.Z.)

**Abstract:** Millet color is an important index for consumers to assess foxtail millet quality. The yellow color of millet is mainly because of the accumulation of carotenoids, which are essential for human nutrition. However, the discoloration of millet during storage due to carotenoid degradation seriously reduces the nutritional and commercial value of millet products. The essential quality traits of millet discoloration during storage were analyzed using two foxtail millet varieties, namely 9806-1 and Baomihunzi. We observed that the millet discoloration was caused by carotenoid degradation during storage, and different genotypes exhibited different discoloration rates. The carotenoid reduction rate in 9806-1 (32.2%) was higher than that in Baomihunzi (10.5%). The positive correlation between carotenoid reduction and the expression of SiLOX protein indicated that SiLOX from foxtail millet played a major role in carotenoid reduction during storage. The expression profiles of the *SiLOX* gene family were analyzed at different grain maturing stages, from S1 to S3, in these two varieties to determine the key *SiLOX* genes responsive to millet discoloration in foxtail millet. The consecutively low expression of *SiLOX2*, *SiLOX3*, and *SiLOX4* contributed to the low level of SiLOX protein in Baomihunzi. Furthermore, the undetectable expression of *SiLOX4* in the later stage of maturation in Baomihunzi was associated with low discoloration, indicating that *SiLOX4* might be a key gene in regulating the discoloration of millet. This study provided critical information on the mechanism of carotenoid degradation during millet storage and laid the foundation for further understanding of carotenoid metabolism in foxtail millet.

**Keywords:** foxtail millet; lipoxygenase; carotenoid; discoloration

## 1. Introduction

Carotenoids are the principal pigments responsible for yellow, orange, and red colors in different plant organs, including flowers, fruits, vegetables, and seeds. They play multiple roles in plant survival, such as acting as light-harvesting antennae in photosynthesis, protecting plants from light damage by quenching triplet-excised states in chlorophyll II, and attracting pollinators and seed distributors in non-photosynthesis processes [1,2]. Carotenoids cannot be synthesized in the human body and must be obtained from the diet. They provide valuable nutrition and health benefits for humans. They produce biologically active molecules that are important antioxidants and free-radical scavengers, which can protect against age-related macular durations and reduce the risk of cancer and cardiovascular diseases [3,4]. The degradation rate is an important contributory factor to the final carotenoid content in plant tissues [5]. The advances in the understanding of carotenoid degradation would be critical for improving the nutritional quality of various plant-based food sources.

Foxtail millet [*Setaria italica* (L.) Beauv.], a minor crop, is essential for food security in semiarid regions of Asia and Africa [6]. Hulled foxtail millet is popular because of its rich

and balanced nutrients. Millet porridge is a favorite daily food for most people in northern China, especially for pregnant and postnatal women [7,8]. Carotenoids, which impart a bright yellow color to millet, are important nutrients in millet [9]. The millet color is the key index to determining foxtail millet quality [10]. However, the discoloration of millet during storage reduces the value and nutritional qualities of millet and millet products, which causes serious economic losses for the millet industry. However, the mechanism underlying the carotenoid degradation in foxtail millet during storage is unclear to date.

Previous studies on wheat, golden rice, and sweet corn have reported that lipoxygenase (LOX) activity was responsible for carotenoid degradation during storage or staleness [11–13]. LOXs are nonheme, iron-containing dioxygenases in plants, animals, and fungi. In plants, they are classified into two major groups 9-LOX and 13-LOX. They play a key role in plant development and defense by various oxidizing lipids, especially through their metabolites, such as jasmonates (JA), green leaf volatiles (GLVs), and death acid, which are involved in response to biotic and abiotic stresses [14]. The *13-LOX TomloxC* is reported to be essential for the synthesis of the flavor volatiles in both tomato fruits and leaves [15]. Similarly, the high expression of two *13-LOXs* (*SbLOX9* and *SbLOX5*) in sorghum-resistant lines induced by the sugarcane aphid infestation and exogenous MeJA treatment predicts the putative function of these genes in the biosynthesis of JA and GLVs [16]. The gene of *cssap92*, encoding a 9-LOX protein, is responsive to *Aspergillu* spp. infection in maize [17]. In addition, 9-LOXs are more involved in the seed qualities related to storage. In rice, 9-LOXs are the major factor influencing seed longevity and viability. Over-expression of *LOX3* in LOX3-normal rice cultivar can accelerate the decrease in seed longevity and germination ability, and the rice seeds with *LOX3* knockout exhibited an improved storability [18,19]. Furthermore, it was reported that *r9-LOX1* was directly responsible for the degradation of carotenoids in golden rice after artificial aging treatment [20]. The deletion of *Lpx-B1* in durum wheat resulted in a 4.5-fold reduction in LOX activity and significant improvement in yellow pigments in pasta [21].

Considering the important function of *LOX* genes on the carotenoid degradation in cereal grains, we investigated the relationship between the LOXs in foxtail millet and the discoloration of millet during storage. For the study, "9806-1" with a fast millet discoloration rate and "Baomihunzi" with a slow millet discoloration rate were selected as research materials. At different storage stages, the total carotenoid contents (TCC) of the two varieties were investigated. The expression patterns of SiLOX protein and *SiLOX* gene family members were compared between the two varieties during grain maturing stages to determine which *SiLOX* gene plays a major role in the discoloration of millet. This study improved the understanding of the molecular mechanism of millet discoloration during storage, which could lay a foundation for developing molecular markers and promoting the molecular breeding of high-quality foxtail millet.

## 2. Materials and Methods

### 2.1. Plant Materials

Two foxtail millet varieties, 9806-1 (Crop Research Institute of the Jilin Academy of Agricultural Sciences, Jilin City, Jilin Province, China) and Baomihunzi (Farming variety of Hulan County, Harbin City, Heilongjiang Province, China), were selected from the core germplasm of foxtail millet in China based on the significant difference in millet discoloration between the two varieties. Maturing grains (S1, initial maturation stage, 14 days of filling stage; S2, mid maturation stage, 21 DOFS; S3, last stage of maturation, 29 DOFS) were collected, hulled, and immediately frozen in liquid nitrogen for RNA and protein isolation [22]. In early October, the foxtail millet was harvested, and the hulled grains of the two varieties were separately kept in Kraft paper bags and stored at room temperature (approximately 23 °C) in the dark. Samples of matured grains from the two varieties were collected at 0, 1, 2, 3, 4, 5, 6, and 7 months during storage for carotenoid extraction analysis.

### 2.2. Experimental Site and Field Experiment

The sowing of the foxtail millets was performed on 1st May, 2020 (Average temperature of 24 °C) at the research station in Shanxi Agriculture University, Taigu, Shanxi province, China (112.31° E, 37.26° N). The study site is located 803.2 m above sea level, and the climate is continental monsoon, warm, and with much rain in the summer. The annual average rainfall is 500–650 mm, and 60% is distributed between June and August. The average temperature range is from a maximum of 25 °C during May and October to an average maximum of −2 °C during November and April of the following year. The crop was sown with row spacing of 0.3 m, plant spacing of 0.2 m, the sowing depth was 5 cm, and the planting plot with 13.5 m$^2$ (4.5 × 3.0 m) for each variety.

### 2.3. The Total Carotenoid Content (TCC) Analysis

The TCC of each variety was analyzed once a month during the storage from November 2020 to June 2021. A modified method based on the version developed by the American Association of Cereal Chemists (AACC) was used to extract carotenoids from grains [10,23]. The harvested grains were freeze-dried for 48 h in a lyophilization chamber guard (−30 °C, 37 pa, CHRIST, Osterode, Germany) and ground into powder (TissueLyser 2, Qiagen, Shanghai, China). Overall, 2 g of millet powder was homogenized in 20 mL water-saturated n-butanol by vortexing for 30 s, followed by being shaken for 3 h at room temperature. The extract was centrifuged at 10,000 g for 10 min at 4 °C (Allegra X-30R, Beckman Colulter, Brea, CA, USA). The supernatant was transferred to a new 50-mL centrifuge tube and diluted to the volume to 25 mL with water-saturated n-butanol. The whole process needed to be protected from light. The absorbance of the extract was measured at 448 nm using a biological spectrometer (BioSpectrometer basic, Eppendorf, Hamburg, Germany). All samples were analyzed in triplicates with three biological replicates.

### 2.4. Chemicals and Solvents

Lutein and zeaxanthin (98% pure, Sigma, St. Louis, MA, USA) were used as standards for quantitative HPLC analysis. Methanol and methyl tert-butyl ether (HPLC-grade, Thermo Fisher Scientific, Shanghai, China) were utilized for chromatographic separations.

### 2.5. HPLC Analysis of Lutein and Zeaxanthin

Carotenoid extracts from samples after storage for 0 and 3 months were separated using a Thermo Fisher HPLC system according to the protocol described previously [22]. Briefly, A C30 column (250 cm × 4.6 mm, 5 mm, YMC, Kyoto, Japan) was maintained at 35 °C. The samples were eluted using a linear mobile phase gradient containing (A) methanol/methyl tert-butyl ether/nanopure water (81:15:4, *v/v/v*) and (B) methyl tert-butyl ether/methanol (90:10, *v/v*) at a flow rate of 1 mL/min, over 30 min. The procedure was as follows: 0–20 min 0%–22.2% B, 20–25 min 22.2%–0% B, 25–30 min 0% B. Lutein and zeaxanthin in the sample extracts were identified through their characteristic absorption spectra and quantified by corresponding peak areas based on the established standard curves.

### 2.6. Western-Blot Analysis

Total proteins from grains at maturing stages (S1, 2, 3) and storage stages (m-0, 7) were extracted. Samples (0.1 g) were ground to form a powder in liquid nitrogen and suspended in 1 mL lysis buffer (2 × SDT lysis buffer, Coolaber, Beijing, China). The extract was homogenized in an ice water bath for 30 min followed by centrifugation (Allegra X-30R, Beckman Colulter, Brea, CA, USA) at 12,000 rpm for 20 min. The protein concentration was quantified using a micronucleus concentration meter (μLite spectrophotometer, BioDrop, Cambridge, UK). Proteins (300 μg) were separated in a 10% SDS-PAGE gel (10% SDS-PAGE Gel Preparation Kit, Coolaber, Beijing, China) and blotted onto nitrocellulose membranes (0.45 nitrocellulose membranes, Coolaber, Beijing, China). The membranes were treated with blocking solution (TBST (10× TBST Buffer, Coolaber, Beijing, China) containing 5% (*w/v*) dried skimmed milk powder) for 1 h and incubated overnight with anti-LOX antibody

in the same solution as the closed solution. Signals were detected using the ECL Western Blotting Analysis System (GE Healthcare, Marlborough, New Zealand). Affinity-purified anti-LOX polyclonal antibody was prepared by Sangon (Anti-LOX rabbit polyclonal antibody, Sangon, Shanghai, China). Peptide sequence from LOX (GVTGKGIPNSTSIC) was synthesized, conjugated with keyhole limpet hemocyanin, and used as an antigen. The antibody against LOX was raised in rabbit antibodyand further purified from immunoblots [24]. The dilution of the LOX antibody for immunoblotting was 1:1000. Quantification was performed using software Image Lab (Bio-Rad Laboratories, Inc., Hercules, CA, USA).

### 2.7. Isolation and Sequence Retrieval of LOX Gene Family from Foxtail Millet

Protein sequences of foxtail millet LOX were obtained from a genome-sequencing library (Phytozome V12.1). LOX sequences from wheat and rice were downloaded from NCBI (http://www.ncbi.nlm.nih.gov/ (accessed on 1 October 2021)). A phylogenetic tree was constructed using Molecular Evolutionary Genetics Analysis (MEGA7) Software (University of Kent, Canterbury, UK) with the Neighbor-Joining (NJ) method having 1000 bootstrap values.

### 2.8. RNA Extraction and Real-Time PCR Analysis

To identify the transcript abundance, total RNA samples were isolated from grains at different maturing stages (S1, 2, and 3) according to the manufacturer's instruction of RNAiso Plus (TaKaRa, Beijing, China). Total RNA was reverse-transcribed using a Takara PrimeScriptTM RT reagent Kit (TaKaRa, Beijing, China) with gDNA Eraser. The synthesized cDNAs were diluted 10 times in ddH2O. Quantitative real-time PCR (qRT-PCR) was performed using the SYBR Premix Ex Taq II (Tli RNaseH Plus, TaKaRa, Beijing, China) Kit in the CFX96™ Real-Time System (Bio-rad, Hercules, CA, USA). The reaction mixture contained 5 μL of SYBR Premix Ex Taq II, 1 μL each of the forward and reverse primers (10 μM), 1 μL of diluted cDNA, and ddH2O to a final volume of 10 μL. The cycling program consisted of 95 °C for 30 s, followed by 40 cycles of 95 °C for 5 s, and 58 °C for 30 s. To confirm the product specificity for each gene primer reaction, a melt curve analysis was performed with continual fluorescence data acquisition during the 60–95 °C melting [25]. Transcript abundance of the housekeeping gene *SiACTIN* was quantified as a reference. At least three biological replicates, each with three replicates, were analyzed for each sample. The primers used in this study are listed in Table 1

**Table 1.** Primer's information for RT-PCR of LOX gene in foxtail millet.

| Gene | Forward Primer Sequence (5′ to 3′) | Reverse Primer Sequence (5′ to 3′) |
| --- | --- | --- |
| *SiACTIN* | TGC TCA GTG GAG GCT CAA CA | CAA GAC ACT GTA CTT GCG CTC |
| *SiLOX1* | AAA TCA CTG GCT GAG GAC ACA T | CAG GAG CTT GAA GAT CGG GT |
| *SiLOX2* | GGT CCT CGG AAA TGT GTT GG | GCT GAA CTT TAC GCA GGC TTA |
| *SiLOX3* | AGG AGT TTG GAC GGG AGA TT | CGAGGTTCCTCTCGATGT |
| *SiLOX4* | CCC TGG AGA TGT CCT CAA AG | ATT GCC GTC CAG ATT TCG |
| *SiLOX5* | GGA CCT AAG GCA GTA TGG A | AGG ATG AAG AGC TTG TTG TT |
| *SiLOX6* | CTG CTG TCC TCG CAC TCC | CTC GCT GTC ATC GTT CCA T |
| *SiLOX7* | GAC CGT CTT CCC TCG CAA | GTC GTC GGG GTA GTA GAT GG |
| *SiLOX8* | CCC AAC AGC GTC ACC ATC TA | CCC CGC CCG AGT ATA ATG AG |
| *SiLOX9* | GCG GGT GAT GAT GAC ATA AGT A | TCT TTG TGG CTA TGA TGA ACG |
| *SiLOX10* | TAC CAC TAC GGC GGC TAC TT | TCT GCG TTG GGA GCA TGT C |
| *SiLOX11* | AAC CTC CTG TCG TCG CAC TC | GGG GTC CTT GTT CCT ACT ATC G |
| *SiLOX12* | TAGAAGCCTACACCGATGATAC | CTTCCAGGTTGTGCTGAATAT |

## 3. Results

### 3.1. Changes in the Content and Composition of Carotenoid in the Two Varieties of Foxtail Millet during Storage

To assess the dynamic profiles of the TCC during the millet storage, two varieties of foxtail millet that exhibited an extreme difference in millet discoloration in our preliminary

experiments were analyzed over a storage period of 7 months. The results revealed that the genotype had a significant impact on the degradation of millet color. As shown in Figure 1a, similar curves were observed in both varieties. However, 9806-1, with a 32.2% discoloration rate, exhibited more discoloration than Baomihunzi (with a discoloration rate of 10.5%). After 1-month storage (M0–M1), the TCC in both varieties was decreased (from 19.23 to 16.60 mg kg$^{-1}$ in 9806-1 and from 11.99 to 11.37 mg kg$^{-1}$ in Baomihunzi). Interestingly, it was noted that the TCC increased at M2 for both varieties. In the following 1-month storage (M1–M2), the TCC in 9806-1 slightly increased to 16.84 mg kg$^{-1}$ and then steadily decreased to 13.03 mg kg$^{-1}$ after five months of storage. However, the TCC in Baomihunzi increased to 12.15 mg kg$^{-1}$ at the M2 stage but decreased to 10.73 mg kg$^{-1}$ after storage for a long duration. Clearly, the significant difference in millet discoloration between both varieties was in the first (M0–M3) period, especially in the first month of storage.

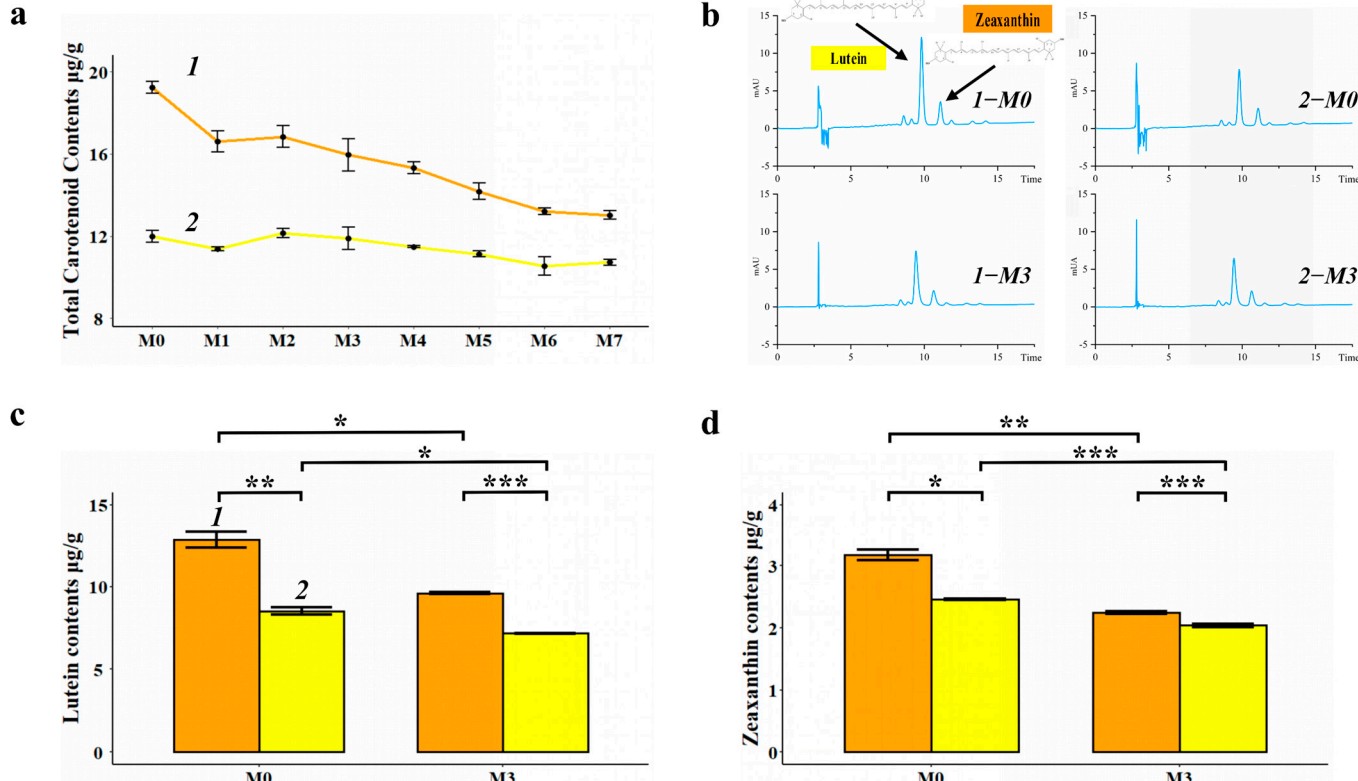

**Figure 1.** Changes in the total carotenoids, lutein, and zeaxanthin contents in the grains of 9806-1 (1, orange) and Baomihunzi (2, yellow) varieties at different storage stages. (**a**) The changes in total carotenoids in the grains of the two varieties during 7-month storage. (**b**) HPLC analysis of carotenoids in two varieties at M0 and M3 stages. Characteristic peaks of lutein and zeaxanthin. (**c**) Comparison of lutein content in the grains of two varieties at M0 and M3 stages. (**d**) Comparison of zeaxanthin content in the grains of two varieties at M0 and M3 stages. The bars represent the standard error (SE) from three biological replicates. Asterisks indicate significant differences; * $p < 0.05$, ** $p < 0.01$, and *** $p < 0.001$.

The YMC C30 carotenoid column was used to isolate carotenoids from these millet samples of the initial (M0) and middle (M3) storage stages; HPLC analysis was performed to quantify changes in the composition of carotenoids during storage. The carotenoid compositions in grains of foxtail millet were comprised mainly of lutein and zeaxanthin (Figure 1b). After three consecutive months of storage, the lutein content decreased from 12.83 to 9.60 mg kg$^{-1}$ in 9806-1 and from 8.52 to 7.17 mg kg$^{-1}$ in Baomihunzi (Figure 1c); however, the zeaxanthin content decreased from 3.18 to 2.24 mg kg$^{-1}$ in 9806-1 and from 2.46 to 2.04 mg kg$^{-1}$ in Baomihunzi (Figure 1d). As expected, the decrease in lutein and

zeaxanthin contents reflected the decrease in the TCC, resulting in the millet discoloration of foxtail millet.

### 3.2. Expression Patterns of SiLOX Protein in the Two Millet Varieties

To investigate if the difference in millet discoloration between the two varieties was related to the expression of SiLOX protein, western blot analysis was performed with polyclonal antibody against LOX. SiLOX protein expression was species-specific (Figure 2), and its expression in Baomihunzi was constantly and significantly lower than that in 9806-1 at all stages of grain maturation and millet storage. It appeared that the higher level of SiLOX protein in 9806-1 was positively related to the higher discoloration rate of carotenoids, in contrast to that in Baomihunzi.

**Figure 2.** Comparison of the expression patterns of SiLOX protein between the grains of 9806-1 and Baomihunzi varieties at three grain development stages and two storage stages.

### 3.3. Phylogenetic Analysis of SiLOX Genes in Foxtail Millet

In total, 12 *SiLOX* genes were identified in foxtail millet by bioinformatic methods. To understand the classification and evolutionary relationship of foxtail millet SiLOXs, a phylogenetic tree was constructed using LOX proteins of foxtail millet and other plants. The SiLOX proteins of foxtail millet can be explicitly categorized into two subfamilies, with seven SiLOXs clustering into the 9-LOX group and five SiLOXs into the 13-LOX group (Figure 3). Phylogenetic analysis revealed that SiLOX2, SiLOX3, SiLOX4, and SiLOX7 were clustered with those of 9-LOXs that were involved in regulating seed longevity and degradation of carotenoids in wheat and rice, indicating that SiLOX2, SiLOX3, SiLOX4, and SiLOX7 may have the same functions.

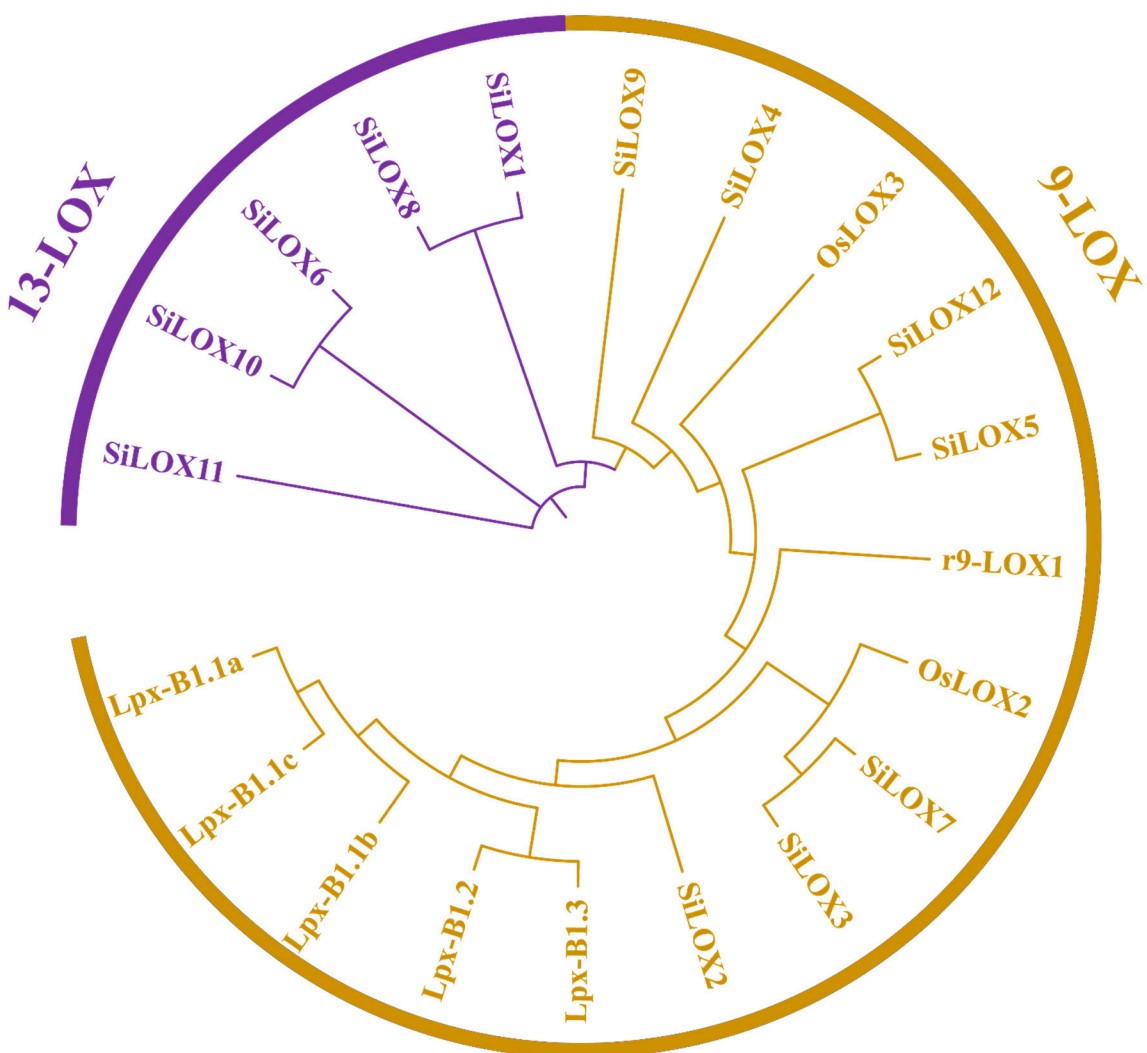

**Figure 3.** Phylogenetic analysis of LOX proteins from *Setaria italica*, *Oryza sativa,* and *Triticum turgidum.* The LOX proteins were divided into two phylogenetic subgroups (9-LOX and 13-LOX) and marked with different line colors.

### 3.4. Tissue-Specific Expression Patterns of SiLOX Genes

To identify the genes related to millet discoloration, the expression profiles of *SiLOXs* in different tissues were analyzed using RT-PCR (Figure 4). The transcripts of *SiLOX1*, *SiLOX5*, and *SiLOX8* could be detected in all the tested tissues. Seven *SiLOX* genes (*SiLOX5*, *SiLOX6*, *SiLOX7*, *SiLOX8*, *SiLOX9*, *SiLOX10* and *SiLOX11*) exhibited high expression levels in leaves, with much higher expression observed for *SiLOX6*, *SiLOX7*, and *SiLOX8*. In addition, *SiLOX10* and *SiLOX11* were highly expressed in seedlings. *SiLOX1*, *SiLOX5*, *SiLOX8*, and *SiLOX9* were strongly expressed in roots. It was noticeable that *SiLOX2*, *SiLOX3*, and *SiLOX4* exhibited stronger expression in grains than in other tissues, especially for *SiLOX3* and *SiLOX4*, which were almost only expressed in grains and their expression was significantly higher. However, *SiLOX9*, *SiLOX10,* and *SiLOX11* were not expressed much in grains. The transcripts of *SiLOX12* could not be detected in any tested tissues.

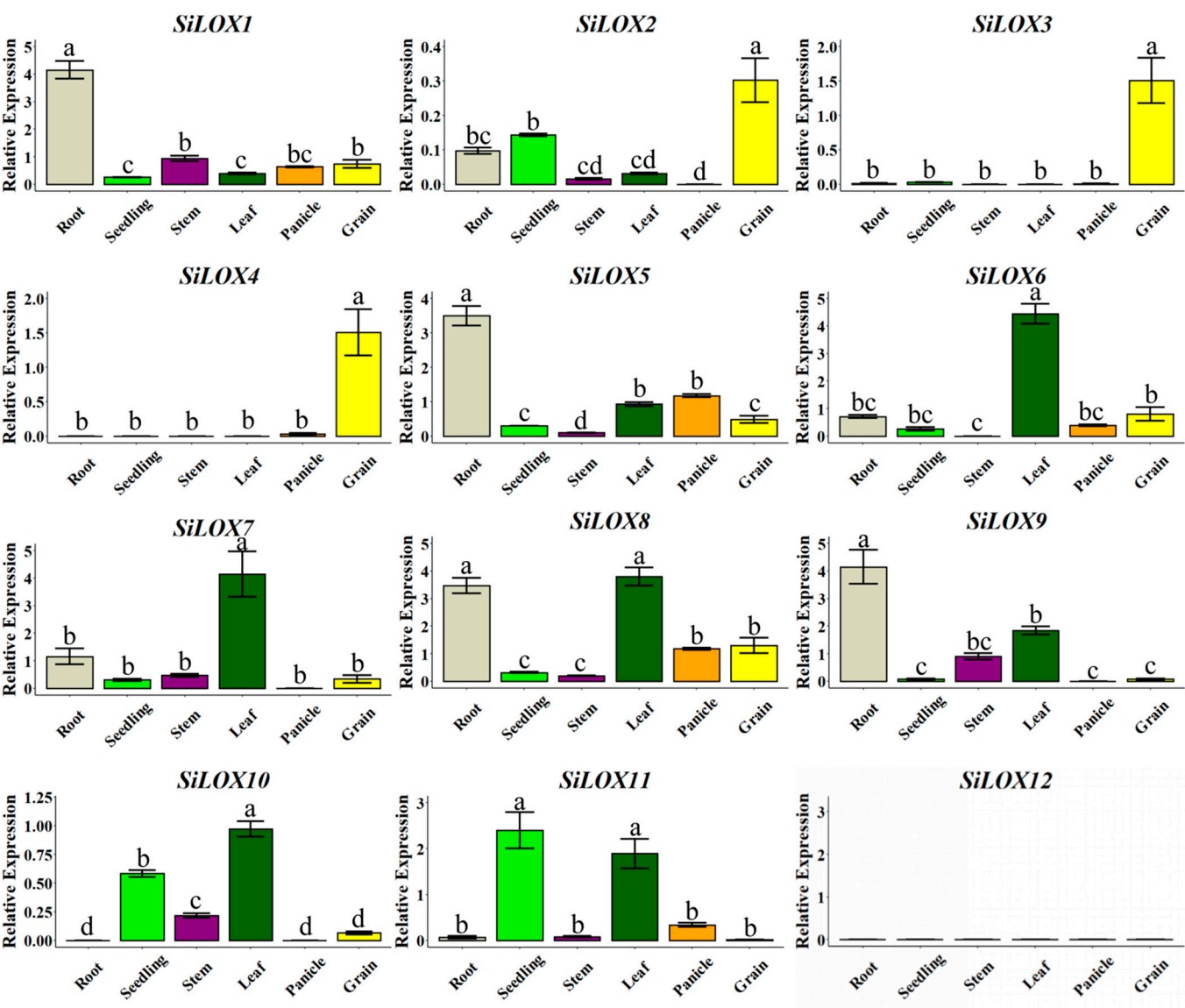

**Figure 4.** The expression profiles of *SiLOX* genes in various tissues of foxtail millet as analyzed by quantitative real-time RT-PCR. The data represent the mean from three replicates with three biological repeats. The bars represent the SE from three biological replicates. The lowercase letters indicated the statistical significance at the level of 0.05 ($p < 0.05$).

### 3.5. Expression Patterns of SiLOX Genes during Grain Maturation

To determine which *SiLOX* gene plays a major role in the discoloration of millet, the expression profiles of the *SiLOX* genes that can be detected in grains were analyzed in two millet varieties during three grain maturation stages (Figure 5). In general, most *SiLOX* genes exhibited higher expression in 9806-1 than in Baomihunzi at all stages, except *SiLOX6* and *SiLOX8*, which exhibited slightly lower expression consistently in 9806-1 than in Baomihunzi. The mRNA levels of *SiLOX1*, *SiLOX5*, and *SiLOX7* exhibited a major increase during grain maturation and were higher in 9806-1 than in Baomihunzi except for *SiLOX5*, which exhibited slightly lower expression in 9806-1 than in Baomihunzi at the S2 stage. The expression of *SiLOX3* kept decreasing during grain maturation in the two varieties and was higher in 9806-1 than Baomihunzi. The expression levels of *SiLOX4* first increased with grain maturation, peaked at the S2 stage, and then decreased dramatically at the S3 stage in both varieties. *SiLOX2*, *SiLOX3*, and *SiLOX4* exhibited significantly higher expression in 9806-1 than in Baomihunzi at the last two stages, especially at the

S3 stage. The expression of *SiLOX2* and *SiLOX3* in 9806-1 was approximately 2.5 times higher than that in Baomihunzi, and the expression of *SiLOX4* could not be detected in Baomihunzi. This indicated that the extremely low expression of *SiLOX2*, *SiLOX3*, and *SiLOX4* contributed to the low level of SiLOX protein in Baomihunzi. Furthermore, it was noteworthy that the expression of *SiLOX4* was undetectable at the last maturing stage.

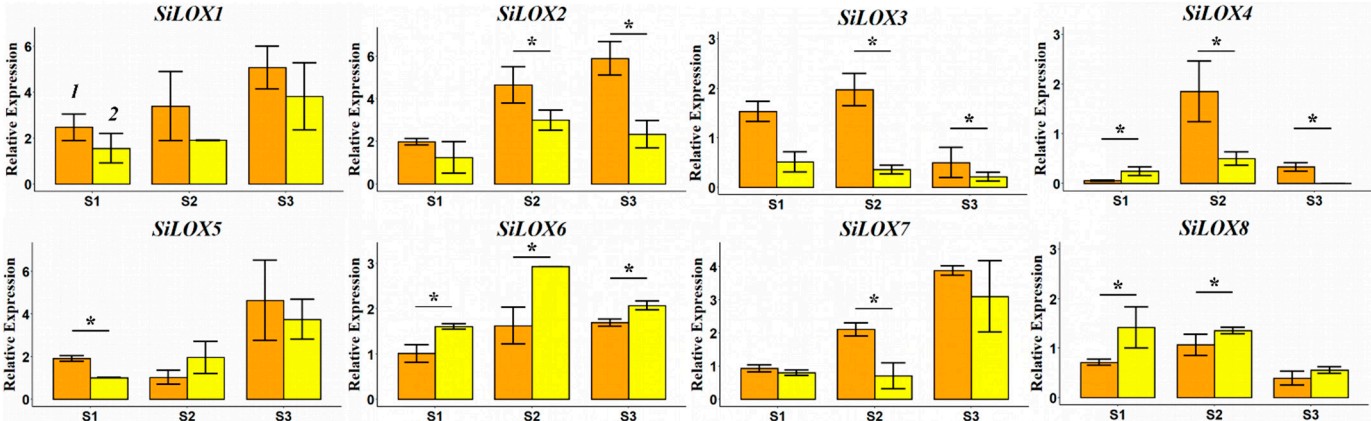

**Figure 5.** Quantitative real-time RT-PCR analysis of the mRNA levels of *SiLOX* genes that can be detected in grains at three grain development stages in 9806-1 (1, orange) and Baomihunzi (2, yellow) varieties. The data represent the mean from three replicates with three biological repeats. The bars represent the SE from three biological replicates. Asterisks indicate significant differences; * $p < 0.05$.

## 4. Discussion

Foxtail millet is an ancient cereal grain crop that is extensively cultured for food in the arid and semiarid regions of China [26]. The primary processing product of the crop is millets. Because of increasing awareness of the health benefits of millet, it is gaining increasing popularity among consumers [10,27]. Carotenoids, a class of important nutrients in foxtail millet, give the millet its yellow color. Unfortunately, the instability of carotenoids due to its high susceptibility to oxidation results in the loss of both color and nutritional qualities of millet products [28]. To date, no information is available on the carotenoid degradation in foxtail millet during storage. In this study, we have explored the potential link between millet discoloration and LOX in foxtail millet. Based on the changes in the TCC in the two millet varieties during storage, it appeared that the genotype had a significant influence on millet discoloration. After long-term storage, the TCC of 9806-1 decreased by 32.25%; however, that of Baomihunzi decreased by only 10.46%. In our prior study, lutein and zeaxanthin were proved to be the two major carotenoid components in millet [22]. Further analysis by HPLC revealed that millet discoloration was mainly caused by the decreasing of lutein and zeaxanthin contents. During the storage of grains in the first month, some water remained in the grains after harvesting. This indicated that the enzymes responsible for the discoloration could be more active; a relatively rapid reduction in the TCC was observed in both varieties. Furthermore, a significant difference in millet discoloration between the two varieties was also observed in M0–M1. The slight increase in the TCC during the following 1-month storage (M1–M2) could be interpreted as the result of regulation by postharvest maturation. The enhancement of carotenoid content in millet during postharvest maturation was commonly observed in other varieties of foxtail millet, and such carotenoid accumulation was accompanied by the starch breakdown in our previous studies. Besides cultivation, the development of carotenoids occurs during postharvest storage [29]. For example, during the storage for the first two months of winter squash, the TCC increased 3-4 times in comparison with that at the time of harvest [30]. Similarly, the analysis of changes in the carotenoid content in mango revealed that while β-carotene and lycopene contents first increased and then decreased, violaxanthin contents

steadily increased during postharvest storage [31]. In our study, during the consecutive last 5 months of storage, although the TCC of both varieties decreased, 9806-1 exhibited a faster degradation rate than Baomihunzi.

Carotenoids are prone to degradation during storage and food production by a lipoxygenase-assisted process. Some evidence has illustrated that LOX was related to the carotenoid degradation during storage or staleness in staple crops [20,21,32]. LOX is a class of nonheme iron-containing dioxygenases that co-oxidize carotenoids by the random attack of the carotenoid molecules [33]. Soybean LOX enzymes are reported to be used as the pigment bleacher to bleach carotenoids in the process of white bread making [34,35]. ZAF1, a maize variety lacking LOX-1and 2, exhibited superior storage stability to other normal varieties [36]. In addition, considerable genetic variability was observed in terms of TCC and LOX activity; genotypes with a high-carotenoid loss may also express low LOX activities [11]. A key constituent involved in carotenoid degradation is endogenous lipoxygenases, and a higher lipoxygenase concentration is associated with an increased loss in lutein content [37]. In this study, the significant difference in the expression of SiLOX protein between the two varieties at all stages supported our hypothesis that SiLOX was involved in the millet discoloration process.

The endogenous lipoxygenase in grain is significantly important in terms of food quality because they have negative implications for color, off-flavor, and nutritional quality. In our previous study, 12 *SiLOX* genes were identified in foxtail millet [38]. Among them, 8 *SiLOX* genes (*SiLOX1-8*) were expressed in the grains of foxtail millet, and the transcript levels of *SiLOX2*, *SiLOX3*, and *SiLOX4* were significantly higher in 9806-1 with a faster discoloration than that in Baomihunzi, especially at the last stage of maturation. *SiLOX2*, *SiLOX3*, and *SiLOX4* were clustered together with *OsLOX3*, *r9-LOX1*, and *Lpx-B1*, which were reported to regulate the storage stability and carotenoid degradation in seeds of rice and wheat [19,21,39]. LOXs in wheat are the best characterized, and their physiological functions are well studied. In durum wheat, the deletion in a lipoxygenase gene, *Lpx-B1.1*, is associated with a strong reduction in LOX activities, leading to improved pasta color [11]. A further study revealed that the expression levels of the *Lpx-B1* gene family contributed to the most total LOX activity, thereby positively correlating with the β-carotene bleaching activities in mature grains [40]. During golden rice storage, the *r9-LOX1* gene plays an important role in carotenoid degradation. After artificial aging treatment, the downregulation of *LOX1* activity by LOX-RNAi can prevent the loss of carotenoids in golden rice seeds [20]. Based on the grain-specific expression characteristic of *SiLOX3* and *SiLOX4* (that they were only expressed in grains of foxtail millet) and reported role of their orthologues in rice and wheat, it was concluded that they might be mainly related to the SiLOX protein expression corresponding to the discoloration of millet during storage. It was noteworthy that the lack of *SiLOX4* expression at the last maturing stage appeared to account for the extremely low expression of SiLOX protein in Baomihunzi, which exhibited slow discoloration. Collectively, our results indicated that *SiLOX4* might be the key factor that affected the millet discoloration rate in foxtail millet during storage.

## 5. Conclusions

In this present study, our objective was to investigate the molecular mechanism of millet discoloration during storage in foxtail millet. Our results suggested that SiLOX played a major role in carotenoid reduction during millet storage. In addition, we found that *SiLOX4* was a key gene in regulating millet discoloration. However, to further determine the function of *SiLOX4* on the carotenoid reduction during storage in foxtail millet, studies with a transgenic approach are necessary.

**Author Contributions:** Conceptualization, B.Z.; Methodology, B.Z.; Formal Analysis, Q.M. and J.W.; Resources, H.L.; Data Curation, Q.M. and L.C.; Writing—Original Draft Preparation, Q.M.; Writing—Review and Editing Y.H., X.Z., and Y.L., Visualization, Q.M. and Q.Z.; Supervision, B.Z. and Y.H.; Project Administration, B.Z.; Funding Acquisition, B.Z. All authors have read and agreed to the published version of the manuscript.

**Funding:** This work was supported by the National Natural Science Foundation of China (grant numbers 31971906, 31601369), Grand Science and Technology Special Project in Shanxi Province (grant numbers 202101140601027), Shanxi Key Laboratory of Minor Crops Germplasm Innovation and Molecular Breeding, Shanxi Agricultural University (grant number 202204010910001), and Shanxi Province Science Found for Excellent Young Scholar (grant number 201901D211382).

**Institutional Review Board Statement:** Not applicable.

**Data Availability Statement:** Not applicable.

**Conflicts of Interest:** The authors declare no conflict of interest.

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
