# Peer review of "The Potential Function of SiLOX4 on Millet Discoloration during Storage in Foxtail Millet"

_agriculture, doi:10.3390/agriculture12081283_

Round 1
Reviewer 1 Report
1. The study is solid. However, there are typos throughout the paper. For example, Bomihunzi vs Baomihunzi and 9806-1 vs 9801-1. Please make sure the term usage is consistent.
2. The title “SiLOX Gene Family is Responsible for the Millet Discoloration during Storage in Foxtail Millet” seems not too precise as the study is based on two selected millets and SiLOX4 is the main contributor for the discoloration.
3. The positive correlation between carotenoid reduction and the expression of SiLOX protein strongly indicated that SiLOX from foxtail millet played a major role in carotenoid reduction during storage (Page 1, Line19-21).
This statement is too strong as based on Figure 5, there is no significant difference data support the comparison between 8 SiLOX. Please include the statistical analysis for the 8 SiLOX during S3.
4. Different genotypes exhibited different discoloration rate. For the study, “9806-1” with fast millet discoloration rate and “Baomihunzi” with slow millet discoloration rate were selected as research materials (Page 2, Line 77).
However, as showed in Figure 1, it indicated that 9806-1 clearly has higher carotenoid than Baomihunzi. Please clarify the reason not to use the two millets have the same amount of carotenoid for research.
5. In western blot analysis, “The extract was homogenized in an ice water bath for 30 min followed by centrifugation at 12,000 rpm for 20 min (Page 3, Line130-131).
Authors should include the centrifuge model or perform the conversion between rpm and g for clarification.
6. Peptide sequence from LOX (GVTGKGIPNSTSIC) was synthesized, conjugated with keyhole limpet hemocyanin, and used as antigen (Page 3, Line138-139).
As showed in Figure 2, western blot Baomihunzi has lower total LOX, please clarify whether or not the LOX, (LOX2, 3, and 4) could be detected when using antibody
7. Its expression in Baomihunzi was constantly and significantly lower than that in 9806-1 at all stages of grain maturation and millet storage. It appeared that the higher level of SiLOX protein in 9806-1 was positively related to the higher discoloration rate of carotenoids, in contrast to that in Baomihunzi (Page 5, Line 206-209).
Can authors descript SiLOX protein function at grain maturation stage. Did it has the same function as at storage stage, or it plays another role during the stage?
8. It was noticeable that SiLOX2, SiLOX3, and SiLOX4 exhibited stronger expression in grains than in other tissues, especially for SiLOX3 and SiLOX4, which were almost only expressed in grains and their expression was significantly higher (Page 7, Line235-238).
Besides SiLOX3 and 4; SiLOX8, 6, and 1 also exhibited stronger expression in grain than SiLOX2 based on Figure 4. Can authors point out which variety and which plant development stage used in Figure 4 -the expression profiles of SiLOX genes in various tissues of foxtail millet?
9. The SiLOX gene expression in grain from Figure 4 did not match those from Figure 5. For example, SiLOX2 in Figure 4 showed lower gene expression than SiLOX 3, 4, 8, 6, 1; but in Figure 5, SlLOX2 has higher expression (Page 8 and Page 9).
Please clarify the discrepancy between the two Figures for SiLOX 2.
Reviewer 2 Report
In the study entitled SiLOX Gene Family is Responsible for the Millet Discoloration 2 during Storage in Foxtail Millet [Setaria italica (L.) Beauv.] authors have tried to improve the understanding of the molecular mechanism of millet discoloration during storage, which may lay a foundation for developing molecular markers and promoting the molecular breeding of high-quality foxtail millet. Further, this study impressed on the mechanism of carotenoid degradation during millet storage and laid the foundation for further understanding of the carotenoid metabolism in foxtail millet.
In the introduction section, the latest literature reference is from 2018, please try to update the recent related literature from last two years to make it more comprehensive and up to date .
Material and methods
Are the two varieties released from the institute of the present study, if not then please provide the source of the seed of these varieties.
Write a few lines on the recommended package of agronomic practices followed while raising the crop and whether is there any correlation between agronomic management ( fertilizer application) and millet discoloration after storage.
Similarly in the discussion section on the claim that SiLOX4 may be the key factor that affected the millet discoloration rate in foxtail millet during storage, provide some evidence based on previous studies in related cereals, if any
Reviewer 3 Report
The results of the manuscript titled "SiLOX Gene Family is Responsible for the Millet Discoloration during Storage in Foxtail Millet [Setaria italica (L.) Beauv.]" written by Qi Ma et al. showed novelty about the possibilities of increasing the nutritional quality of plant products. It is now very important to carry out such experiments. I highlight that the experiment was conducted in field conditions. The manuscript has a well-constructed Introduction chapter and also a Discussion chapter in which the results obtained are widely communicated with other authors. Nevertheless, I have some well-meaning comments about the manuscript that could increase its quality and develop some ideas:
1. The research site could be defined by geographic coordinates.
2. All used material (including biological material), machines and devices must be specified (production name, company name, city, country of origin).
3. All used methods must be specified and supported by citations.
4. I regret to note that the manuscript lacks a more detailed description of the setting up of the field experiment and the agrotechnical operations used. I recommend that the new subchapters entitled "Experimental site" and "Field experiment" could be added to the Materials and Methods chapter.
5. In the case of a field experiment focused on carotenoids, the results may be significantly affected by the interaction of the weather conditions and soil properties. In Materials and Methods chapter, I kindly ask the authors for a detailed course of weather (temperature and precipitation) of individual months of the experimental period. In the manuscript, this can be supplemented by graphs (course of weather) or tables (soil properties) to help understand some of the differences in results achieved. Information could be added to the "Experimental site" subchapter.
6. The manuscript lacks a "Conclusion" chapter. I kindly request the authors to incorporate in the conclusion the idea of using the results obtained in practice in the scope of agronomy and human nutrition.
Reviewer 4 Report
Ma et al. have investigated SiLOX Gene Family and its responsibility for the Millet discoloration during Storage in Foxtail Millet.
In general, the manuscript contains relevant paragraphs that have been discussed. The selection of the bibliography is appropriate to the content of the manuscript.
Moreover, regarding methodology, the authors refer to statistics. Thus, the readers can make assumptions regarding the quality and the confidence of the results and the reasonability of consideration of the authors.
In conclusion, the manuscript is enjoyable to read, but after a close evaluation of the paper, I suggest some corrections.
1- There are minor punctuation and English corrections in the attached file
2- Is any reference confirming the difference in millet discoloration between the two studied varieties?
3- In figure 1, the author has to mention the meaning of every color
4- Why do the authors not use LOX cDNAs besides LOX proteins in studying the Phylogenetic Analysis of SiLOX Genes?
5- There are no asterisks above the SE bars in figure 4
6- In figure 5, the author has to mention the meaning of every color and what meaning of S1, S2, and S3 in the figure legend
7- Why do the authors not do a correlation analysis between Changes in the total carotenoids, lutein, and zeaxanthin contents and the gene expression analysis?
Best Regards

Round 2
Reviewer 3 Report
Dear authors, thank you for incorporating most of my comments. The manuscript is now better prepared than when I first reviewed it. In your cover letter, you state that points 2 and 4 from my first review have been incorporated into the manuscript. However, I regret to say that this is not the case. Having incorporated these two points, I recommend the manuscript for publication. I wish you every success in your research.
Author Response
Response to Reviewer 3 Comments
Point 2: All used material (including biological material), machines and devices must be specified (production name, company name, city, country of origin).
Response 2: Thank you for the suggestions. We have added the relevant materials and equipment information as required. Latest version is highlighted in red.
Point 4: I regret to note that the manuscript lacks a more detailed description of the setting up of the field experiment and the agrotechnical operations used. I recommend that the new subchapters entitled "Experimental site" and "Field experiment" could be added to the Materials and Methods chapter.
Response 4: Thank you for the suggestions. We have added "Experimental site and Field experiment" part in the “Materials and Methods” section. Latest version is highlighted in red.
